# Using machine learning to determine the shared and unique risk factors for marijuana use among child-welfare versus community adolescents

**Sonya Negriff** [1‡]*, **Bistra Dilkina** [2‡], **Laksh Matai** [2], **Eric Rice** [3]

**1** Department of Research and Evaluation, Kaiser Permanente Southern California, Pasadena, California, United States of America, **2** Department of Computer Science, University of Southern California, Los Angeles, California, United States of America, **3** Suzanne Dworak-Peck School of Social Work, University of Southern California, Los Angeles, California, United States of America

‡ SN and BD are contributed equally to this work as co-first authors.
* sonya.x.negriff@kp.org

**Data Availability Statement:** There are ethical restrictions on sharing a de-identified dataset because they contain sensitive information about child welfare involvement and maltreatment

## Abstract

### Objective

This study used machine learning (ML) to test an empirically derived set of risk factors for marijuana use. Models were built separately for child welfare (CW) and non-CW adolescents in order to compare the variables selected as important features/risk factors.

### Method

Data were from a Time 4 ($M_{age}$ = 18.22) of longitudinal study of the effects of maltreatment on adolescent development (n = 350; CW = 222; non-CW = 128; 56%male). Marijuana use in the past 12 months (none versus any) was obtained from a single item self-report. Risk factors entered into the model included mental health, parent/family social support, peer risk behavior, self-reported risk behavior, self-esteem, and self-reported adversities (e.g., abuse, neglect, witnessing family violence or community violence).

### Results

The ML approaches indicated 80% accuracy in predicting marijuana use in the CW group and 85% accuracy in the non-CW group. In addition, the top features differed for the CW and non-CW groups with peer marijuana use emerging as the most important risk factor for CW youth, whereas externalizing behavior was the most important for the non-CW group. The most important common risk factor between group was gender, with males having higher risk.

### Conclusions

This is the first study to examine the shared and unique risk factors for marijuana use for CW and non-CW youth using a machine learning approach. The results support our

experiences. Sharing a de-identified dataset on this small vulnerable group could potentially lead to identifiable information given the location of the study is specified in the manuscript and the dataset includes age, race, and gender. Requests may be sent to Julie Cederbaum PhD, MSW, the lead of the data access committee at USC School of Social Work, at jcederba@usc.edu.

**Funding:** This study was funded by National Institutes of Health Grants R01HD39129 and R01DA024569 (to P.K. Trickett., Principal Investigator). The funders had no role in study design, data collection and analysis, decision to publish, or preparation of the manuscript.

**Competing interests:** The authors have declared that no competing interests exist.

**Abbreviations:** AUC, area under the curve; CPS, children and family services; CW, child welfare; IPV, intimate partner violence; ML, machine learning; PFI, Permutation Feature Importance; ROC, receiver operating curve; SVM, Support Vector Machines.

assertion that there may be similar risk factors for both groups, but there are also risks unique to each population. Therefore, risk factors derived from normative populations may not have the same importance when used for CW youth. These differences should be considered in clinical practice when assessing risk for substance use among adolescents.

## Introduction

Child maltreatment and subsequent involvement with child welfare (CW) is a significant issue that affects a large number of youth in the US [1]. In addition to the wide-ranging effects on physical and mental health [2–5], child maltreatment is a known risk for alcohol, marijuana and illicit drug use in adolescence and adulthood [6–11]. While not all children who are maltreated come to the attention of CW, those who do enter CW have higher rates of substance use than the general population [12, 13], highlighting the need to understand the specific risk factors in this vulnerable population. In addition, adolescence is a key developmental period for prevention, as data indicate 90% of adults who meet the criteria for addiction, initiated use of alcohol or drugs in adolescence [14–16]. Identifying the risk factors for early substance use among adolescents will help prevent future abuse and alleviate the economic toll of substance use/abuse [17–20].

### Adolescence is a key developmental period for the initiation of substance use

The confluence of biological, social, and cognitive changes that occur during adolescence increase the potential risk for initiation and prolonged use of substances [21]. Foremost, brain development during adolescence is primed for risk-taking behaviors, with executive function in the prefrontal cortex lagging behind the increased growth of the reward and sensation seeking regions [21]. Evidence also indicates that the adolescent brain is more sensitive to the addictive properties of nicotine, alcohol and other drugs, increasing the propensity for addiction [22, 23]. Social influences also increase vulnerability; susceptibility to peer pressure peaks in mid-adolescence and peer substance use is a known risk factor for substance use [24]. Early timing of puberty is also associated with higher risk of substance use through initiation of sexual behavior and exposure to older peers [25]. According to the Centers for Disease Control and Prevention, 90% of adults who meet the criteria for addiction initiated use of alcohol or drugs in adolescence [14–16], highlighting this developmental period as a key time for prevention. More specifically, evidence indicates that individuals who initiate marijuana use in early adolescence are more likely to be prolonged users and progress to marijuana dependence [26]. Similarly, early alcohol misuse has been linked with abuse in adulthood. The effects of substance use in adolescence range from injury [27], unintended pregnancy [28], mental health problems [29, 30], impaired brain function [31–33], reduced academic performance [34, 35], and criminal involvement[36]. The economic toll of substance use/abuse is estimated to be over $740 billion annually as a result of accidents, health care, homelessness, unemployment, and criminal activity [17–20].

### Predictors of early substance use for youth involved with child welfare

There is a substantial body of evidence addressing factors that predict alcohol, marijuana, and illicit drug use among community samples of adolescents [37], yet the relative contributions of

these risk factors to substance use for CW youth have largely been examined with samples comprised only of CW-involved youth (no comparison group) [38]. In the only known comparison of CW and non-CW youth, Fettes and colleagues [12, 39] used the National Longitudinal Study on Adolescent Health and the National Survey of Child and Adolescent Well-Being (NSCAW) to examine the known risks for marijuana, inhalant, and other illicit drug use between CW and non-CW- involved youth. Results showed that parental closeness and parental education predicted current substance use among CW youth but not the community sample, and two-parent household predicted lifetime and current use among the community sample [39]. These findings demonstrate that the expected risk factors do not operate similarly for CW youth, necessitating further work to delineate the relative importance of known predictors of substance use for CW versus non-CW adolescents.

## Machine learning predictive models for early substance use among high risk youth

The development of more accurate predictive analytics can provide new opportunities for the early detection of high-risk youth that can go beyond the identification of broad epidemiologic type categories such as gender, race, and CW involvement. There has been an increased interest and much commentary in developing predictive models that use Machine Learning (ML) to help hone intervention efforts in medicine [40–45], substance abuse [46], and child welfare [47]. It has been shown in various domains that the broader class of machine learning techniques is often able to achieve more accurate predictive models than conventional statistical models such as logistic regression. A significant advantage of ML methods is the ability to analyze large amounts of data and to uncover relationships that remain hidden to standard statistical technique by allowing for more complex multi-way relationships between dependent and independent variables [48]. In addition, established strategies in machine learning, such as train/test split and cross-validation help with model calibration, prevent over-fitting, and measure performance in terms of generalization to unseen test data. Importantly, ML can advance our understanding of the predictors of early substance use using both data-driven and theory-driven selection of risk factors and can accommodate more potential predictors than conventional statistical techniques.

A number of studies have applied ML approaches to enhance our understanding of substance use, abuse, and relapse [46, 49–52]. However, the majority have used adult sample with fewer focused on adolescents. In a study of Australian and Canadian adolescents four distinct clusters of predictors (demographics, psychopathology, risk behaviors, personality) were used as predictors of alcohol use, with personality and psychopathology clusters yielding the highest prediction accuracy indices [53]. Individually, sensation seeking, attention problems, prior alcohol use, and negative thinking were among the features with the highest predictive coefficients. A cohort study of nearly 700 adolescent included six features (demographics, family history, genetics, brain images, personality, and cognition) to predict current and future alcohol drinking [54]. Overall, stressful life events were found to contribute the most unique variance to the predictive model. While evidence suggests the utility of ML models for alcohol, cocaine, heroin, and substance use disorders, as of yet ML has not been applied to investigate marijuana use among adolescents.

## The current study

Some studies [12, 39] have examined the same risk factors for substance use among CW-involved and non-CW-involved youth, albeit using different studies, and others have examined the factors specific to CW youth [38]. However, there are no studies delineating both the

shared and unique risks for substance use among CW versus community youth within the same study. The extant evidence indicates the established risks for early substance use for community youth are not the same for CW youth. Therefore, further research is needed to pinpoint the risk factors specific to CW youth and those that may be shared across both CW and community youth. The current study used machine learning to test an empirically and theoretically derived set of risk factors for marijuana use. Models were built separately for CW and non-CW adolescents in order to compare the variables selected as important features/risk factors. The findings will enhance our understanding of the relative importance of risk variables among CW-involved youth to provide better information for screening into services.

## Methods

### Participants

Data were from the fourth assessment (T4) of an ongoing longitudinal study examining the effects of maltreatment on adolescent development. At Time 1 (T1), the sample was composed of 454 adolescents aged 9–13 years (241 males and 213 females). Time 2 (T2), Time 3 (T3), and Time 4 (T4) occurred on average 1, 2.7, and 7.2 years after baseline. Descriptives of the sample for baseline (T1) and T4 can be found in Table 1.

**Recruitment.** The participants in the child welfare group (*N* = 303) were recruited from active cases in the Children and Family Services (CFS) of a large west coast city. The inclusion criteria were: (1) a new referral to CFS in the preceding month for any type of maltreatment (e.g. neglect, physical abuse, sexual abuse, emotional abuse); (2) child age of 9–12 years (some turned 13 between scheduling and actual study visit); (3) child identified as Latino, African-American, or Caucasian (non-Latino); (4) child residing in one of 10 zip codes in a designated county at the time of referral to CFS. With the approval of CFS and the Institutional Review Board of the University of Southern California, caregivers of potential participants were contacted via postcard and asked to indicate their willingness to participate. Contact via mail was followed up by a phone call. Of the families referred by CFS, 77% agreed to participate.

The non-child welfare group (N = 151) was recruited using names from school lists of children aged 9–12 years residing in the same 10 zip codes as the maltreated sample. Caregivers of

**Table 1. Sample characteristics for Time 1 and 4.**

| | Child welfare | | Non-Child welfare | |
|---|---|---|---|---|
| | **Time 1** | **Time 4** | **Time 1** | **Time 4** |
| N | 303 | 222 | 151 | 128 |
| Age (std deviation) | 10.84 (1.15) | 18.28 (1.41) | 11.11 (1.15) | 18.15 (1.56) |
| Gender (%) | | | | |
| Male | 50 | 47 | 60 | 56 |
| Female | 50 | 53 | 40 | 44 |
| Ethnicity (%) | | | | |
| African American | 40 | 43 | 32 | 35 |
| Latino | 35 | 34 | 47 | 42 |
| White | 12 | 10 | 10 | 10 |
| Multi-racial | 13 | 13 | 11 | 13 |
| Living Arrangement (%) | | | | |
| With Parent | 52 | 56 | 93 | 85 |
| Foster Care or Extended Family | 48 | 24 | 7 | 3 |
| Without caregiver | n/a | 20 | n/a | 12 |
| Marijuana use (% ever used) | — | 48.2 | — | 41.4 |

potential participants were sent a postcard and asked to indicate their interest in participating which was followed up by a phone call. Non-CW families confirmed that they had no previous or ongoing experience with child welfare agencies. Approximately 50% of the comparison families contacted agreed to participate.

Upon enrollment in the study the CW and non-CW groups were compared on a number of demographic variables (see Table 1). The two groups were similar on age, (CW *M* = 10.84 years, *SD* = 1.15; non-CW *M* = 11.11, *SD* = 1.15), gender (53% male), race (38% African American, 39% Latino, 12% Multi-racial, and 11% Caucasian), and neighborhood characteristics (low-income based on Census block information) [reference withheld for blind review]. However, they were different in terms of living arrangements. In the non-CW group 93% lived with a biological parent, whereas this was the case for only 52% of the CW group. The remainder of the CW group was living in foster care, which is not unusual for those adolescents involved with social services.

**Retention.**   The retention rate between T1 and T4 was 77.5% (n = 352). Participants not seen at Time 4 were more likely to be in the CW group (OR = 2.45, *p* < .01) and male (OR = 1.86, *p* < .01).

## Procedures

Assessments were conducted at an urban research university. After written consent from the caregiver and assent from the adolescent was obtained, they each completed the questionnaires and tasks in separate rooms. The measures used in the following analyses represent a subset of the questionnaires administered during the protocol. Both the child and caretaker were given remuneration compatible with National Institutes of Health's standard compensation rate for healthy volunteers. The Institutional Review Board of the University of Southern California reviewed and approved all study procedures.

## Measures

**Outcome.**   *Marijuana use*. Participants reported on their own marijuana use within the past 12 months via one item from the Adolescent Delinquency Questionnaire [ADQ; adapted from [55]]. Due to the needs of the prediction model, the number of times the adolescent used marijuana (0 to five or more) was re-coded as 0 (no use) or 1 (any use).

**Risk factors.**   *Demographics*. Age was calculated from date of birth and date of interview (continuous), gender was given by the parent at enrollment in the study (male vs. female), and race/ethnicity (African American, Hispanic, Multi-racial and White) was reported by the parent using a demographic questionnaire. Race/ethnicity used as four separate variables indicating that particular race/ethnicity versus all others.

*Mental health symptoms*. Symptoms of depression, anxiety, and post-traumatic stress were included as risk factors. Adolescents completed the 27-item Children's Depression Inventory. [56, 57] They rated statements such as "I am sad all the time" and "I feel like crying every day," on a three-point scale, with the total score used in the analyses (range of possible scores = 0–54). The Cronbach's alpha for T4 was .89. Symptoms of PTSD occurring in the past couple of months were assessed using the Youth Symptom Survey Checklist (Margolin G. The Youth Symptom Survey Checklist. Los Angeles, CA: Unpublished manuscript; 2000). This is a 17-item self-report measure of symptoms from the diagnostic criteria for PTSD found in the Diagnostic and Statistics Manual of Mental Disorders IV-TR such as hyperarousal, avoidance/ numbness, and re-experiencing. Answer options range from 1 = not at all to 4 = almost always. The total score was used for this analysis (17 items; $\alpha$ = .88) and can range from 17 to 68. The 39-item Multidimensional Anxiety Scale for Children [58] was used to measure anxiety

symptoms. It has been found to have good internal consistency (range for subscales is .70–.89), good test-retest reliability, invariant factor structure across gender and age, and discriminant validity [58]. The nine items on the separation anxiety subscale (e.g., "I get scared when my parents go away") were removed from the scale at T4 due to development inappropriateness. Items such as "I feel tense or uptight" were rated from 0 to 3 ("never true about me" to "often true about me") yielding a possible total score range from 0–90. Internal consistency reliability was .89 at T4.

*Self-reported childhood maltreatment and adversities (self-reported ACEs)*. The Comprehensive Trauma Interview (CTI) [59] was used at Time 4 to assess self-reported exposure to maltreatment and adversities. The CTI assesses 19 different adverse experiences including parental divorce, parental incarceration, witnessing intimate partner violence (IPV), household substance use, death of parent, foster care placement or other parental separation, sexual abuse, physical abuse, emotional abuse, emotional neglect, and physical neglect. The CTI was administered via interview by a trained research assistant. Other studies have shown test-retest reliability ranging from .45-.76 depending on the maltreatment type [60, 61]. For the current analyses we used 7 individual items: witnessing IPV, household substance use, sexual abuse, physical abuse, emotional abuse, emotional neglect, and physical neglect (each coded 0 = no or 1 = yes). Community violence exposure was assessed with 19 items asking about witnessing violence "in your neighborhood or around your school" [62]. Items included "in the past year have you seen a person beaten up without a weapon" on a scale from never = 0 to more than 8 times = 4. The items were summed to create a composite score for exposure.

*Risk behavior*. Sexual behavior was measured using the Sexual Activity Questionnaire for Girls and Boys [63]. This questionnaire assesses series of eleven sexual activities with a current boyfriend/girlfriend as well as a past partner or with anyone. Activities begin with holding hands, continue with kissing, heavy petting, and culminate in sexual intercourse. The eleven sexual behavior items were summed (no = 0, yes = 1) to create a composite score of sexual behavior with higher scores indicating more advanced sexual behavior. Alcohol use was measured using the ADQ with one item indicating "how many times in the past 12 months have you passed out drunk". The Youth Self Report was used to measure externalizing behavior [64]. The externalizing subscale is composed of aggression (17 items) and rule-breaking/delinquency (12 items). Each item is rated from 0 to 2 ("not at all" to "a lot") with a possible range of 0–58. Cronbach's alpha was .89 at T4. The participants also reported on their own delinquent behaviors within the past 12 months via 23 items from the Adolescent Delinquency Questionnaire (ADQ; adapted from [55]. Computerized administration was used to ensure participant confidentiality. For the present study three scales were used: status offenses (6 items, e.g. "run away from home", $\alpha$ = .72-.74), person offenses (7 items, e.g. "carried a hidden weapon", $\alpha$ = .77-.83), and property offenses (10 items, e.g. "damaged or destroyed someone else's property on purpose", $\alpha$ = .88-.92). The three scales were summed to create a composite score for delinquency.

*Peer risk behavior*. Participants reported on the delinquency and substance of their peers within the past 12 months. Similar to the adolescent self-report, they were asked "how many of your friends or people your age you know have done this in the past 12 months". Answer options were 0 = none, 1 = some, 2 = a lot. Three scales of delinquency were used (status offences 6 items, $\alpha$ = .72-.80, person offences 7 items, $\alpha$ = .77–82, property offences 10 items, $\alpha$ = .85-.90) and summed to create a composite score for peer delinquency. One item was used to assess peer marijuana use and one to assess peer alcohol use ("how many of your friends or people your age you know have used marijuana in the past 12 months" and "how many of your friends or people your age you know have had an alcoholic drink in the past 12 months"). The same answer options were used as with the delinquency items.

*Parent/family social support*. The Hill intimacy scale was included to assess the degree of intimacy/closeness with parents [65]. The adolescent is asked to answer about their mother or the person who acted most like your mother, and a second time about their father, or father figure. There are eight questions such as "how much do you go to your mother for advice or support" answered on a three point scale: none, some, or a lot. The Cronbach's alpha for the mother scale was .85 and for the father was .91. The two scores were averaged, or if missing one then only that one was used. Parental monitoring was assessed using the AddHealth Parental Monitoring questions [66]. This scale is comprised of six questions such as "How often do you tell your parent(s) who you were going out with?" rated on a scale from 0 = never to 4 = always. The total score was used in analyses and the Cronbach's alpha was .82 at T4.

*Self-esteem*. Two components of self-esteem were used, global self-worth and self-image [67]. The Self-Perception Profile for Adolescents (SPPA) [68] is a widely used self-report measure assessing global self-worth (synonymous with self-esteem)[69–71]. The current study collected six of the original eight subscales: athletic competence, scholastic competence, social competence, behavioral conduct, self-acceptance, and close friendship. The scales were summed in order to create a composite score of global self-worth (α = .85). Self-image was assessed using two subscales (*body image and mastery/coping)* of the Self-Image Questionnaire for Young Adolescents (SIQYA) [72]. This self-report measure is designed for children ages 11 to 15 years and these subscales were selected because of their particular relevance to adolescents. The total sum scale had an internal consistency reliability of *α* = .85.

## Data analysis

The small amount of missing values in the dataset (see Table 2) was addressed using multiple imputation (MI). Although the percent missingness was below 2% for each individual variable, listwise deletion (as required by our ML models) would have resulted in dropping 10% of the total sample. As such, we determined that MI was necessary to account for potential bias in missingness. Five imputed datasets were created using the MI function in SPSS 25.0.

Following MI, all non-binary predictor variables were standardized to allow for better interpretability of the coefficients. This was followed by a multi-step analysis plan to establish the best performing ML model and to identify the top features contributing to its performance. It should be noted that in the context of ML, the terms "variables" and "features" are used interchangeably.

Our analytic approach used traditional statistical methods (binary logistic regression) as well as ML approaches: Lasso and Support Vector Machines (SVM). Logistic Regression is a well-known and often used statistical technique for binary classification [73] that is easy to interpret but has limited model capacity and suffers from overfitting when many features are considered. Lasso regularization often helps in overcoming overfitting [74]. It achieves this by adding a penalty of the absolute value of the magnitude of the coefficients. Support Vector Machines [75] are widely used and very successful models for classification. SVM interprets the predictor values for each data point (i.e., person in our study) as a vector of coordinates in p-dimensional space and searches for a (p-1)-dimensional hyperplane in that space that separates the points belonging to the two classes with the largest margin or gap possible. In linear SVM, one can interpret the magnitude and sign of the coefficient in the linear hyperplane similarly to the coefficients in Logistic Regression and Lasso [76]. All ML analyses were performed in Python using the package Scikit-learn [77]. We used AUC (area under the curve [AUC] of the receiver operating characteristic [ROC], or C-statistic) as our main metric of goodness of fit, where generally a value higher than 0.7 designates a good model, and higher than 0.8 a strong model [78]. The Receiver Operating Characteristic (ROC) curve shows the tradeoff

**Table 2. Individual predictor variables (features), domains, and descriptives.**

| Feature Domain | Feature name | Coding | Range | % missingness | CW: Bivariate correlation with MJ use | nonCW: Bivariate correlation with MJ use |
|---|---|---|---|---|---|---|
| Demographics | Race | white = 0 minority = 1 | [0, 1] | none | .096 | .230** |
| Demographics | Gender | female = 0 male = 1 | [0, 1] | none | -.168* | -.166 |
| Demographics | Age | | [14.71, 22.66] | none | .154* | .016 |
| Mental health | Anxiety | continuous | [0, 85] | 0.57% | -.097 | .033 |
| Mental health | Depression | continuous | [2, 40] | 0.57% | .070 | .146 |
| Mental health | PTSD | continuous | [17, 66] | 1.70% | .050 | .317** |
| Parent/family social support | Parental closeness | continuous (low = risk) | [1,3] | none | .023 | -.297** |
| Parent/family social support | Parental monitoring | continuous (low = risk) | [0, 24] | none | -.258** | -.158 |
| Parent/family social support | Social support | continuous (low = risk) | [1,5] | 0.85% | -.092 | -.196 |
| Peer risk behavior | Peer delinquency | continuous | [0, 46] | none | .286** | .250** |
| Peer risk behavior | Peer alcohol use | 0 = none 1 = some 2 = a lot | [0,1,2] | 1.14% | .203** | .129 |
| Peer risk behavior | Peer marijuana use | 0 = none 1 = some 2 = a lot | [0,1,2] | 1.14% | .380** | .232** |
| Risk behavior | Sexual activity | score 0–11 | [0, 11] | 1.99% | .261** | .265** |
| Risk behavior | Delinquency | continuous | [0, 30.5] | none | .085 | .384** |
| Risk behavior | Externalizing | continuous | [0,36] | 1.70% | .165* | .461** |
| Self-esteem | Global self-worth | continuous (low = risk) | [8,20] | none | -.077 | -.215* |
| Self-esteem | Self-image | continuous (low = risk) | [38, 126] | 1.99% | .017 | -.093 |
| Self-report ACEs | Emotional abuse | no = 0; yes = 1 | [0, 1] | 1.42% | .025 | .303** |
| Self-report ACEs | Emotional neglect | no = 0; yes = 1 | [0, 1] | 1.42% | .079 | .167 |
| Self-report ACEs | Household substance use | no = 0; yes = 1 | [0, 1] | 1.42% | .038 | .095 |
| Self-report ACEs | Physical abuse | no = 0; yes = 1 | [0, 1] | 1.42% | .122 | .306** |
| Self-report ACEs | Physical neglect | no = 0; yes = 1 | [0, 1] | 1.42% | -.083 | .207* |
| Self-report ACEs | Sexual abuse | no = 0; yes = 1 | [0, 1] | 1.42% | .131 | .219* |
| Self-report ACEs | Witnessing IPV | no = 0; yes = 1 | [0, 1] | 1.42% | .076 | .098 |
| Self-report ACEs | Witnessing community violence | continuous | [0, 32] | 1.14% | .247** | .330** |

Note: CW = child welfare; ACEs = adverse childhood experiences; MJ = marijuana; IPV = intimate partner violence.

**$p < .01$,

*$p < .05$

between true positives (sensitivity) and false positives (1-specificity) at all possible thresholds, and hence the area under the ROC curve measures the overall accuracy of the model without choosing a specific threshold. In addition, we also report precision and recall at the threshold of 0.5. Precision measures the fraction of the youth who were indeed marijuana users among those predicted to use marijuana, while recall (sensitivity) measures the fraction of all youth who used marijuana that the model actually predicted as such. For each of the three approaches (logistic regression, Lasso, SVM), we performed the following steps.

First, due to the large number of potential features, we performed feature selection to remove redundant features that might degrade the model performance. We used a technique called Backward Feature Selection [79] which iteratively selects to remove a feature/variable starting with all features, and evaluating the predictive performance with a single feature removed and choosing the one whose removal results in the best AUC. Given this selection, then we again re-evaluate the performance when removing one of the remaining features and select the best one, based on AUC. This process stops when only one feature remains. The method returns the number and names of the features that resulted in the best AUC over these iterative steps.

Next, k-fold cross-validation was used to evaluate the model performance. This technique of validation makes sure that every data point was once part of the test samples and alleviates possible sensitivity to selecting a single split of the data into a training and a test subset. It does this by splitting the data into k-groups. For each unique group, the group is held as a test set while the others combined are used for training the model. In the end, the model's performance on each test set is retained and the final score is the average performance across all the k-groups. Since we have 352 participants, the value of k was chosen to be 5 for all experiments, which means 70 participants were part of the test set at each validation.

To determine which features contributed most strongly to the model performance we performed Permutation Feature Importance (PFI) analysis [80, 81]. Permutation Feature Importance is a widely used technique for calculating feature importance that it is model-agnostic, i.e., it works for any of the predictive approaches [80, 81]. It randomly permutates a single feature/predictor in the validation dataset leaving all the other features intact and then computing the AUC of the model on this permutated validation set. The PFI value of a feature is the respective drop in AUC observed. The larger the decrease from the original AUC, the higher the rank importance of the feature. Following PFI, we averaged (across k = 5 folds) the linear coefficients for those features with PFI $\geq 0.005$ ($\geq .5\%$ drop in AUC) in order to understand the positive or negative effect of a feature on the predicted likelihood of marijuana use. Finally, for each of the three approaches, we computed the average k-fold AUC across the five imputations to determine the most accurate model for each of the two groups (CW and non-CW).

We performed sensitivity analyses to determine if there were differences between the original (non-imputed) and imputed data on the performance metrics for the ML models. The AUC was the essentially the same for the both the CW and non-CW groups in the non-imputed data and imputed across all three models (see S1 Table). Therefore, we report only the results from the imputed data.

## Results

### Descriptives

As shown in Table 1, 48.2% of the CW group and 41.4% of the non-CW group reported marijuana use in the past year. Bivariate correlations between the predictors and marijuana use for each group (Table 2) showed that in the CW youth, marijuana use was positively associated with peer delinquency, peer alcohol use, peer marijuana use, sexual activity, externalizing behavior, and witnessing community violence ($p < .05$). Additionally, there were negative associations of marijuana use with gender (female) and parental monitoring ($p < .05$). For the non-CW youth, marijuana use was positively correlated with race (minority), PTSD, peer delinquency, peer marijuana use, sexual activity, delinquency, externalizing, emotional abuse, physical abuse, physical neglect, sexual abuse, and witnessing community violence ($p < .05$). It was also negatively correlated with parental closeness, social support, and self-esteem competence ($p < .05$).

**Table 3. Performance metrics for the three machine learning approaches.**

| | CW | | | Non-CW | | |
|---|---|---|---|---|---|---|
| | **AUC** | **Precision** | **Recall** | **AUC** | **Precision** | **Recall** |
| Logistic Regression | 0.79 ± 0.004 | 0.72 ± 0.013 | 0.73 ± 0.009 | 0.87 ± 0.028 | 0.72 ± 0.003 | 0.68 ± 0.037 |
| Lasso | 0.80 ± 0.001 | 0.71 ± 0.015 | 0.75 ± 0.018 | 0.85 ± 0.021 | 0.72 ± 0.007 | 0.66 ± 0.014 |
| SVM | 0.80 ± 0.012 | 0.72 ± 0.01 | 0.74 ± 0.025 | 0.84 ± 0.010 | 0.73 ± 0.011 | 0.69 ± 0.057 |

Note: ± indicates the range across the 5 imputed datasets.

## Machine learning models

**CW group.** The performance metrics for each of the three ML approaches can be found in Table 3. For the CW group, the AUC for all three models was very similar and indicated good model fit (logistic regression AUC = .79, Lasso AUC = .80, SVM AUC = .80). In addition, we also report precision and recall at the threshold of 0.5. Like AUC, precision and recall for all three models was very similar (see Table 3). As an example, the logistic regression model achieved 0.72 precision (the fraction of the youth who were indeed marijuana users among those predicted to use marijuana), while recall was 0.73 (the fraction of all youth who used marijuana that the model actually predicted as such). Because the AUC, precision, and recall for the three ML models were not appreciably different, we could not choose one model as superior over the others. Therefore, feature selection and permutation feature importance (PFI) were performed on all three and we retained only those features that met the PFI threshold of 0.005 across all three models. This strategy increases the robustness of the feature selection as it mitigates model specific uncertainty. Using these criteria, eight features were retained (Fig 1a). The top feature was peer marijuana use which reduced the AUC by 12–13% (across the three models) if dropped from the model. Other features in order of importance were parental monitoring, gender, sexual abuse, age, physical neglect, witnessing community violence, and parental closeness. The linear coefficients produced by the model (Fig 2a) indicated that six features were risk factors whereas parental monitoring and physical neglect were protective. Specifically, higher levels of peer marijuana use, male sex, self-reported sexual abuse, older age, self-reported physical neglect, witnessing community violence, and higher levels of parental closeness all predicted marijuana use. On the other hand, higher levels of parental monitoring and self-reported physical neglect predicted non-use.

**Non-CW group.** The AUC for all three models was very similar and indicated good model fit (logistic regression AUC = .87, Lasso AUC = .85, SVM AUC = .84) and slightly better accuracy than the CW model. At threshold of 0.5, the logistic regression model for non-CW achieved 0.72 precision (the fraction of the youth who were indeed marijuana users among those predicted to use marijuana), while recall was 0.68 (the fraction of all youth who used marijuana that the model actually predicted as such). Again, because the model performance metrics were very similar across all three we chose to retain only those features that achieved our cut-off of .005 for PFI across all three models. This resulted in four features being retained (Fig 1b). In order of importance these were: externalizing behavior (9–12% drop in AUC if removed from the model), delinquency, gender, and parental closeness. The sign of the linear coefficients indicated that higher externalizing behavior, higher delinquency, higher peer marijuana use, and male gender were all risk factors for marijuana use while, higher levels of parental closeness was protective (Fig 2b).

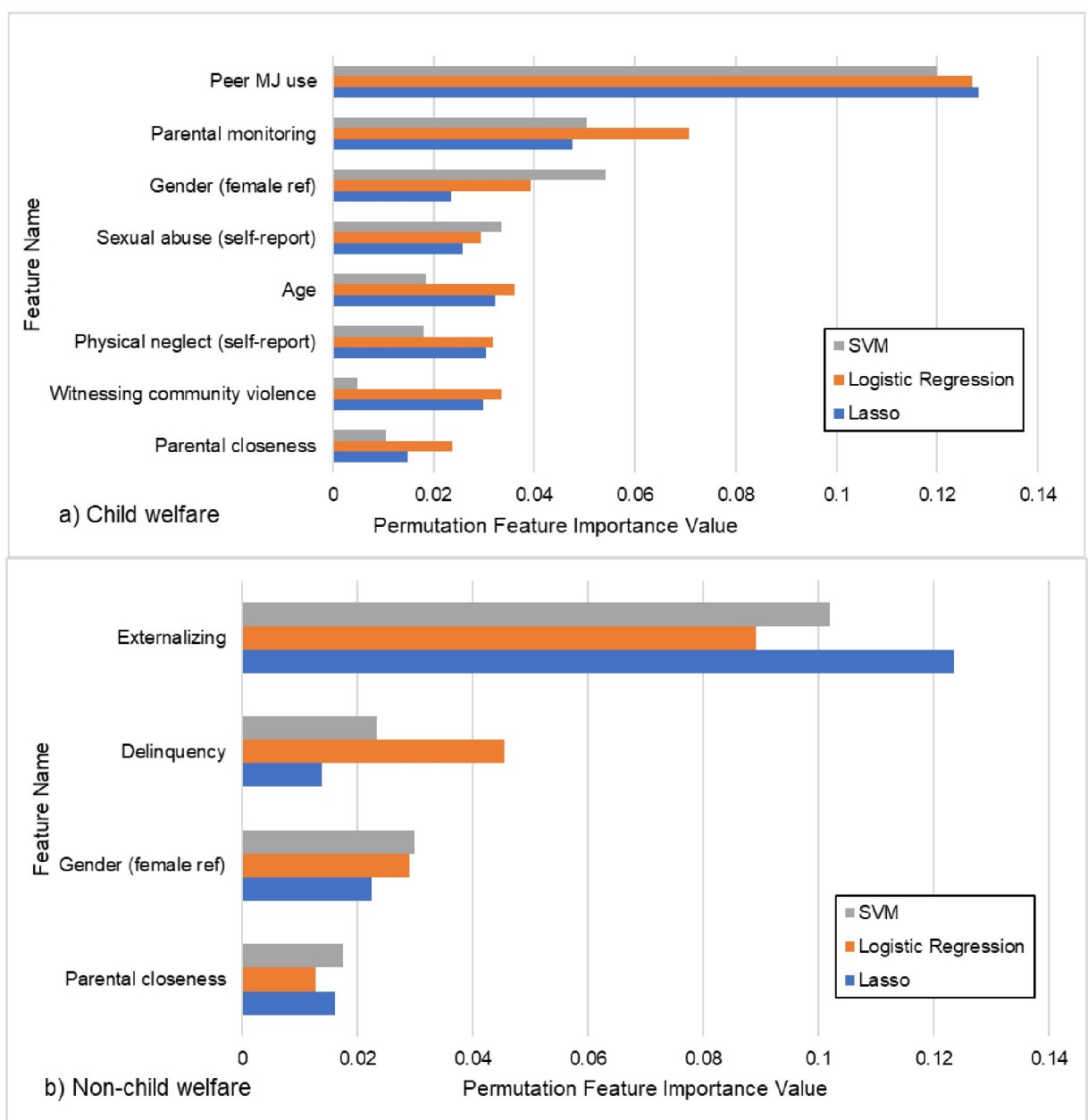

**Fig 1. Plot of individual predictors selected by model ranked by Permutation Feature Importance value for a) Child Welfare and b) non-Child Welfare groups.**

## Discussion

The risks for substance use in adolescence have been extensively studied on normative populations, however CW youth are a particularly vulnerable population and may have unique risks. Our results support this supposition by showing that although the CW and non-CW groups shared some key predictors of marijuana use, they also had a substantial number of unique key predictors including a different top ranked predictor. These findings demonstrate the importance of developing separate predictive models for these different populations of youth and

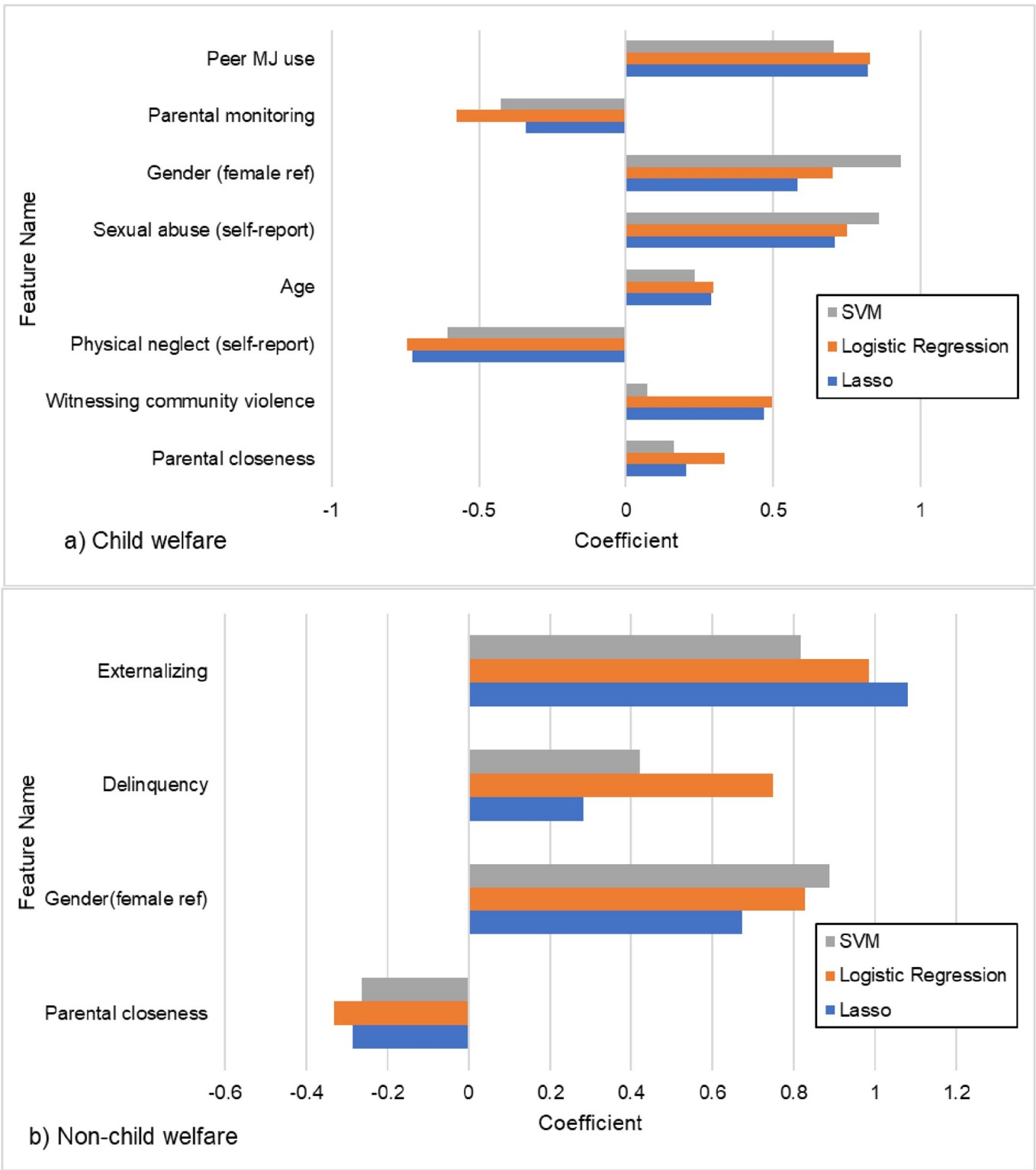

**Fig 2. Plot of individual predictors selected by model ranked by coefficient for a) Child Welfare and b) non-Child Welfare groups.**

that generalizing results from normative populations may miss key predictors of marijuana use for CW youth.

## Shared predictors for CW and non-CW groups

In an attempt to increase the robustness of the feature selection and reduce model-specific idiosyncrasies, we only retained those features that met our cut-off for PFI in all three models. Using this strategy, eight features were retained for the CW youth while four were retained for the non-CW youth. Of those features, only two were shared by both groups: male gender and parental closeness. While males have been shown to have higher risk for marijuana use in both normative [82–84] and CW samples [38], parental closeness had opposite associations in CW versus non-CW group. Higher parental closeness was a risk for the CW youth but protective for the non-CW youth. The expectation was that parental closeness would be protective in both groups [84, 85], yet as others have shown expected associations do not always hold in youth with maltreatment or CW experiences [86]. Maltreated youth are more likely to experience insecure attachments with their parents [87] and although they report closeness, it may be reflective of an unhealthy attachment style which may exacerbate vulnerability for risk behavior such as marijuana use. Future studies might consider how in certain context close family relationships may actually be detrimental, as in the case where parents might be abusing substances and modeling this behavior for the adolescent [88].

## Differences between CW and non-CW groups

The top two features of importance for the CW group were peer marijuana use and parental monitoring, while for non-CW youth they were externalizing problems and delinquency. All of these variables have been found to be consistent predictors of adolescent substance use among both CW and non-CW adolescents with parental monitoring being protective [38, 82, 83]. Interestingly, although externalizing and delinquency were included in the initial set of variables considered by the model for both groups, they were not retained in the final set of important features for the CW group. This result implies that behavior problems are not important for predicting marijuana use for CW youth and peer behavior may be more critical to assess. This contradicts findings from the NSCAW data where delinquency was a risk factor [89] and Aarons et al (2008) where externalizing problems was a predictor of substance use for CW youth. Our results may diverge in part because we specifically tested marijuana use as our outcome rather than a combination of alcohol, marijuana and hard drugs or because of other nuances of our study design. Importantly, because of our use of machine learning we were able to include far more predictors in the model at one time than prior studies. This may yield different associations with outcomes as the relative importance of each variable is assessed in combination with all others, rather than just in a small subset of possible risk factors.

Our results also indicate that CW youth's marijuana use was influenced by older age, self-reported sexual abuse, self-reported physical neglect, and witnessing community violence, Evidence supports these variables to be risks [11, 90–93]. However, the negative coefficient for physical neglect is unexpected, which may be an artifact of the particular combination of variables in the model and potential collinearity between predictors. As evidence accumulates on predictors of substance use among CW youth, it is clear that no consistent pattern of risks has emerged. In a meta-analysis of studies examining predictors of substance use among current and former foster youth, over 15 different variables emerged as predictors across studies [38]. While the strength of our machine learning approach is the ability to test a large number of potential variables without concern for error associated with multiple statistical tests, it is clear that more work needs to be done with larger datasets to converge on a clear set of risk factors.

## Limitations

A limitation of this study is that we did not examine CW-specific predictors that have been shown to be important for substance abuse risk (e.g., placement history, number of referrals, caregiver type). For example, foster care placement is associated with five times higher risk for substance abuse compared to youth who were allowed to remain in their home of origin [94]. We did not include these variables because we were trying to keep the predictors the same across both groups. An additional limitation is that we used a binary variable for our outcome, use/no use. This does not allow examination of the full range of potential use, since 1 use in the past 12 months is combined with those using 5 or more times. However, the identification of early initiation of marijuana use is clinically important for prevention of future substance abuse. We examined only concurrent predictors for marijuana use, it is possible the predictive model may change if data from earlier timepoints was included. We chose concurrent data as this is likely the available data in practice setting and will be useful in terms of assessing current risk for marijuana use. Our predictive modeling strategy is limited in the ability to infer causal or explanatory relationships, instead it uses concurrent information to predict whether a given adolescent is a marijuana user. Finally, it is possible that unreported maltreatment may have occurred in our non-CW group. This is mitigated by our definition of our groups based on child welfare involvement and we do not suggest generalizing the results to individuals who may have maltreatment experiences but were not reported to child welfare.

## Conclusions

Substance use is a substantial public health problem, especially among CW youth. Our study is only the second to provide evidence regarding the comparability of risk factors for substance use among CW-involved versus non-CW-involved youth, but the first to so within one study and using machine learning approaches. Our findings show that while there was some overlap in the most important risks for marijuana use among CW and non-CW youth, the order of importance differed with peer marijuana use emerging as the top feature for CW and externalizing behaviors for non-CW. In addition, more features were retained in the CW model than the non-CW model implying a more complex interplay of risk factors is needed to accurately predict marijuana use for CW youth. The results also support our assertion there are shared risk factors, but also features unique to each population. Therefore, risk factors derived from normative populations will not have the same predictive power when used for CW youth. These differences should be considered in clinical practice when assessing risk for substance use among high-risk adolescents.

## Supporting information

**S1 Table. Performance metrics for the three machine learning approaches for non-imputed (raw) data.**
(DOCX)

## Author Contributions

**Conceptualization:** Sonya Negriff, Bistra Dilkina, Eric Rice.

**Data curation:** Sonya Negriff, Laksh Matai.

**Formal analysis:** Sonya Negriff, Bistra Dilkina, Laksh Matai.

**Investigation:** Sonya Negriff, Bistra Dilkina, Eric Rice.

**Supervision:** Sonya Negriff, Bistra Dilkina, Eric Rice.

**Validation:** Bistra Dilkina.

**Writing – original draft:** Sonya Negriff, Bistra Dilkina, Laksh Matai.

**Writing – review & editing:** Sonya Negriff, Bistra Dilkina, Laksh Matai, Eric Rice.

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
