## [Decision Letter · Decision Letter 0]

17 Feb 2022

PONE-D-21-15650

Using machine learning to determine the shared and unique risk factors for marijuana use among child-welfare versus community adolescents

PLOS ONE

Dear Dr. Negriff,

Thank you for submitting your manuscript to PLOS ONE. After careful consideration, we feel that it has merit but does not fully meet PLOS ONE’s publication criteria as it currently stands. Therefore, we invite you to submit a revised version of the manuscript that addresses the points raised during the review process.

I apologize for the unusually long time that the review process took, but I needed to secure good reviews both from the theoretical and methodological points of view. Both reviewers see potential in your manuscript and I agree, but there are several aspects that require improvement. Nonetheless, they don't identify unfixable flaws, hence, I am confident you can address all their comments. I will send your review to the same reviewers.

We look forward to receiving your revised manuscript.

Kind regards,

Carlos Andres Trujillo, PhD

Academic Editor

PLOS ONE

Journal Requirements:

2. Thank you for stating the following financial disclosure: "Funding acknowledgements: National Institutes of Health Grants R01HD39129 and R01DA024569 (to P.K. Trickett., Principal Investigator)."

5. We note that you have referenced "Margolin G. The Youth Symptom Survey Checklist. Los Angeles, CA: Unpublished manuscript; 2000" which has currently not yet been accepted for publication. Please remove this from your References and amend this to state in the body of your manuscript: "Margolin G. The Youth Symptom Survey Checklist. Los Angeles, CA: Unpublished manuscript; 2000" as detailed online in our guide for authors

http://journals.plos.org/plosone/s/submission-guidelines#loc-reference-style.

Reviewers' comments:

Reviewer's Responses to Questions

**Comments to the Author**

1. Is the manuscript technically sound, and do the data support the conclusions?

Reviewer #1: Yes

Reviewer #2: Partly

2. Has the statistical analysis been performed appropriately and rigorously? 

Reviewer #1: Yes

Reviewer #2: No

3. Have the authors made all data underlying the findings in their manuscript fully available?

Reviewer #1: No

Reviewer #2: Yes

4. Is the manuscript presented in an intelligible fashion and written in standard English?

Reviewer #1: Yes

Reviewer #2: Yes

5. Review Comments to the Author

Reviewer #1: The study proposes a statistical novel approach to understand the interplay of risk factors of marijuana use, comparing a child welfare population and a non-welfare population. The authors argue that there is little research addressing the specific risk factors that affect welfare children, and that they could differ from community children. The manuscript is well written and explains clearly step by step the LM which helps understand the process for the analysis

When the introduce variables such as depression, anxiety and PTSD as risk factors, the age of onset of such psychological issues should be addressed, because those problems could be the result of drug use (and marijuana not being the onset drug it may be more probable), then they will no be risk factors, but outcomes. If the age of onset is not clear or the authors do not have it, then it should be addressed in the limitations section.

The researchers included the use of alcohol in their variables, but it is more abuse than use, which is different. They address the alcohol use by asking how many times in the past 12 months have your passed out drunk. Even if the answer is never, the adolescent may be drinking a lot, but he is not getting drunk, or he could have been drunk but not necessarily passed out. It is clear why they have dichotomized the dependent variable (marijuana use), because of the model they are using, but it should also be reviewed what happen when the variable is maintained with its full scope of categories.

For better understanding, it is important also to explain how was measured peer drug use. The authors said it was measured through one item, but they do not explain how. It is important that at the introduction, the researchers clarify if they are really measuring self-esteem, or they are just measuring competence and body image. They should support this notion of self-esteem by literature that argues that this is self-esteem, and not self-efficacy or just self-competence. It is also important to discuss why this kind of variables have no impact on the dependent variable in either of the two groups.

It is really interesting the use of ML to analyze the risk and protective factors, because it reveals new ways in which the risk and protective factors interact. The study of this factors through LM can be an ideal analytic approach to studying multiple variables such as risk and protective factors. The study has several strengths, including its data-driven approach to study many factors that could influence marijuana use in both samples. One of the things that I could question about the paper is the way use of marijuana was measured. The possible answers were some, a lot, none. I think this type of categorization can diminish the possible variance that could be found in this kind of behaviors. It is also difficult to understand what the difference could be, for the adolescent, between a lot and some use.

It is really surprising the result that parental closeness is a risk factor for children in welfare. Even though the authors give some hypothesis, it could also be interesting to explore, or to propose for future research if, maybe based on the social development model (Hawkins & Catalano), the family, in certain circumstances could become a risk factor, because the child or adolescent has an affective attachment, so she follows her family behaviors, and parents could be using drugs themselves. It could also be a way the adolescent reacts depending on her placement. The authors, in their limitations said that they did not account for placement as a variable so they could pair both samples, but maybe that is one of the reasons of this result. I am not sure that the dichotomization of the race variable is accurate.

Another surprising result is that peer delinquency reduces marijuana use in community sample. I the researchers suggest it was a variable that did have issues in the weight it has, but I think that the problem could be how the adolescent sees his peers, or the dichotomization of the dependent variable. The authors should explore a little farther this result.

Finally, I don’t agree that black and Latino should necessarily share the same backgrounds or contextual variables.

Reviewer #2: PLoS One Review:

Using machine learning to determine the shared and unique risk factors for marijuana use among child-welfare versus community adolescents

1. I suggest more extension and detail in how decision trees work in ML techniques. No reference in lines 56-58: “Not only can ML provide potentially more accurate predictive models, but techniques such as decision trees can provide new insights into non-linear relationships.”.

2. Lines 82-83: “there are no studies delineating both the shared and unique risks” is the italic necessary?

3. Why did CW=child welfare and non-CW turn into Line 117: maltreated and comparison groups? Please have uniformity.

4. What does a “biracial” participant’s race mean?

5. According to data analysis, why do not present results before and after multiple imputations? The missingness proportion is less than 2%, but if you apply multiple imputations, results sure present how data and results look before imputation.

6. Here is one serious concern with the analytical approach: Lines 242-243 describe how “generally a value higher than 0.7 designates a good model, and higher than 0.8 a strong model”. Table 3 presents AUC for all models between 0.79 and 0.83. That means all models (i.e., Logistic Regression, Lasso and SVM) are good or strong. Why do you opt for different models under small and marginal differences?

7. Following the previous comment in Line 237- 239: In linear SVM, one can interpret the magnitude and sign of the coefficient in the linear hyperplane similarly to the coefficients in Logistic Regression and Lasso. What references do you have for this affirmation? Also, it is unclear whether your study is a comparison (e.g., CW=child welfare and non-CW), but you might use non-comparable models (e.g., Logistic Regression vs SVM). Unusually, two different models are recognized for two independent samples under the same analysis. For CW, it is SVM, and for N-CW, it is the Logistic Regression. It is necessary to have a better explanation and justify with technical background why a model may be more applicable to one group and not to the other. This point needs a deep rationale for your data analysis proposal.

8. Line 272 It is “perturbed” or “permuted”?

9. Why do you use the restrictions of the Lasso model and the technique called Backward Feature Selection? It seems like there are too many restrictions for the exploration of Line 46: “further work to delineate the relative importance of known predictors of substance use /…/”.

10. How sensitive is the Permutation Feature Importance (PFI) to the SVM and the logistic regression results?

11. It is not easy to interpret the values from the Permutation Feature Importance (PFI) analysis. However, if the arbitrary selection of models we asked about in point 7 is justifiable, at least all results in all models should be presented for a fair comparison.

12. Since the authors claim that this is a study “Line 396-398: to provide evidence regarding the comparability of risk factors for substance use among CW-involved versus non-CW-involved youth, but the first to so within one study and using machine learning approaches.”, it is capital to clarify the analytic concerns mentioned before. The richness of their approach will reside in how well the points 5 to 11 are corrected and rationalized.

6. PLOS authors have the option to publish the peer review history of their article (what does this mean?). If published, this will include your full peer review and any attached files.

Reviewer #1: No

Reviewer #2: **Yes: **Juan J Giraldo-Huertas

---

## [Author Response · Author response to Decision Letter 0]

21 Jul 2022

RESPONSE TO REVIEWERS

We thank the editor and reviewers for their time and thoughtful comments. We have addressed all comments below and indicated where changes have been made in the manuscript. 

Editor’s comments

RESPONSE: We have made sure that we followed the PLOS ONE style requirements.

2. Thank you for stating the following financial disclosure: "Funding acknowledgements: National Institutes of Health Grants R01HD39129 and R01DA024569 (to P.K. Trickett., Principal Investigator)."

RESPONSE: We have revised the funding statement in the manuscript to note the role of the funders. We have also included this in the cover letter.

RESPONSE: The data contain sensitive information about child welfare involvement and maltreatment experiences. Sharing de-identified dataset on this small vulnerable group could potentially lead to identifiable information given the location of the study is specified in the manuscript and the dataset includes age, race, and gender. Requests may be sent to the lead of the data access committee at USC School of Social Work, Julie Cederbaum PhD, MSW. We have included this information in the cover letter.

RESPONSE: We have included the ethics statement at the end of the procedures section. We already stated at the beginning of the procedures section that written consent was obtained from the caregiver and assent from the adolescent. 

5. We note that you have referenced "Margolin G. The Youth Symptom Survey Checklist. Los Angeles, CA: Unpublished manuscript; 2000" which has currently not yet been accepted for publication. Please remove this from your References and amend this to state in the body of your manuscript: "Margolin G. The Youth Symptom Survey Checklist. Los Angeles, CA: Unpublished manuscript; 2000" as detailed online in our guide for authors

http://journals.plos.org/plosone/s/submission-guidelines#loc-reference-style.

RESPONSE: We have updated this reference as requested.

Reviewer #1: 

The study proposes a statistical novel approach to understand the interplay of risk factors of marijuana use, comparing a child welfare population and a non-welfare population. The authors argue that there is little research addressing the specific risk factors that affect welfare children, and that they could differ from community children. The manuscript is well written and explains clearly step by step the LM which helps understand the process for the analysis

RESPONSE: We thank the reviewer for their positive appraisal of our manuscript.

When the introduce variables such as depression, anxiety and PTSD as risk factors, the age of onset of such psychological issues should be addressed, because those problems could be the result of drug use (and marijuana not being the onset drug it may be more probable), then they will not be risk factors, but outcomes. If the age of onset is not clear or the authors do not have it, then it should be addressed in the limitations section.

RESPONSE: We are using variables concurrent with our outcome to determine if information from the same timepoint can predict whether an adolescent will be a marijuana user. This is different than using these variables as potential causal factors which would necessitate that they precede the outcome. In predictive models we are not attempting to use the “predictor variables” as explanatory or causal features. Instead, the predictive model attempts to use the available information on other variables to make a prediction about the likelihood that a given adolescent is a marijuana user or not. In this type of model it does not matter if the drug use or mental health symptoms occurred first, it only matters if concurrently having mental health symptoms allows us to predict the outcome (marijuana use) with higher accuracy. We have added the limitation of this approach in the discussion.

The researchers included the use of alcohol in their variables, but it is more abuse than use, which is different. They address the alcohol use by asking how many times in the past 12 months have your passed out drunk. Even if the answer is never, the adolescent may be drinking a lot, but he is not getting drunk, or he could have been drunk but not necessarily passed out. RESPONSE: We thank the review for this comment. We agree that alcohol use and our variable “passed out drunk” are very different. We had to choose between including more introductory alcohol use or more severe abuse. If we used alcohol use the rates were very high and gave us little variability. This is in part why we chose the passed out drunk variable.

It is clear why they have dichotomized the dependent variable (marijuana use), because of the model they are using, but it should also be reviewed what happen when the variable is maintained with its full scope of categories.

RESPONSE: To maintain a categorical variable we would need to use different Machine Learning Models and this would not allow for comparison with the models we have already reported on. In addition, the dataset is too small to use 5 categories within the marijuana use variable, the cell sizes would limit the ability to predict the outcome. There are also clinical implications for using a dichotomous variable of use/no use, as any use in adolescence should be of concern and warrant further assessment or intervention.

For better understanding, it is important also to explain how was measured peer drug use. The authors said it was measured through one item, but they do not explain how. 

RESPONSE: We have added clarification of these variables in the “Peer risk behavior” section of the measures.

It is important that at the introduction, the researchers clarify if they are really measuring self-esteem, or they are just measuring competence and body image. They should support this notion of self-esteem by literature that argues that this is self-esteem, and not self-efficacy or just self-competence. It is also important to discuss why this kind of variables have no impact on the dependent variable in either of the two groups.

RESPONSE: We appreciate the opportunity to clarify these issues. In the Self-esteem section of the measures, the two questionnaires we included both measure aspects of self-esteem. Global self-worth is a construct synonymous with self-esteem and self-image is viewed as a component of self-esteem. We have included references to support this conceptualization. It is also not that these variables have no impact, but that other variables in the model are more important in predicting marijuana use. As seen in the bivariate correlations, higher global self-worth is protective for marijuana use.

It is really interesting the use of ML to analyze the risk and protective factors, because it reveals new ways in which the risk and protective factors interact. The study of this factors through LM can be an ideal analytic approach to studying multiple variables such as risk and protective factors.

RESPONSE: We thank the reviewer for highlighting this new way to analyze substance use in adolescence.

The study has several strengths, including its data-driven approach to study many factors that could influence marijuana use in both samples. One of the things that I could question about the paper is the way use of marijuana was measured. The possible answers were some, a lot, none. I think this type of categorization can diminish the possible variance that could be found in this kind of behaviors. It is also difficult to understand what the difference could be, for the adolescent, between a lot and some use.

RESPONSE: We thank the reviewer for raising this issue so that we can clarify. The outcome variable for marijuana use was dichotomized as ‘no use’ versus ‘any use’. It might be that the reviewer is referring to how the peer marijuana use was coded. The three answers referred to how many of their friends were using marijuana. 

It is really surprising the result that parental closeness is a risk factor for children in welfare. Even though the authors give some hypothesis, it could also be interesting to explore, or to propose for future research if, maybe based on the social development model (Hawkins & Catalano), the family, in certain circumstances could become a risk factor, because the child or adolescent has an affective attachment, so she follows her family behaviors, and parents could be using drugs themselves. It could also be a way the adolescent reacts depending on her placement. The authors, in their limitations said that they did not account for placement as a variable so they could pair both samples, but maybe that is one of the reasons of this result. 

RESPONSE: We thank the reviewer for this suggestion. We have added in the discussion section: “Future studies might consider how in certain context close family relationships may actually be detrimental, as in the case where parents might be abusing substances and modeling this behavior for the adolescent.”

I am not sure that the dichotomization of the race variable is accurate.

RESPONSE: We used this dichotomization because evidence indicates higher substance use rates among minority youth. However, we have completed new analyses with race included individually for White, Black, Hispanic, and Multi-racial. We did not find any of these variables to be important predictors of marijuana use. We now report on these findings.

Another surprising result is that peer delinquency reduces marijuana use in community sample. I the researchers suggest it was a variable that did have issues in the weight it has, but I think that the problem could be how the adolescent sees his peers, or the dichotomization of the dependent variable. The authors should explore a little farther this result.

RESPONSE: We agree that this was a surprising finding. However, in our updated analyses this does not emerge as an important feature and has been removed from the results. 

Finally, I don’t agree that black and Latino should necessarily share the same backgrounds or contextual variables.

RESPONSE: We assume this is a reference to the race/ethnicity groups we created (white versus minority youth). If so, then we have addressed this by including individual race/ethnicity variable.

Reviewer 2

1. I suggest more extension and detail in how decision trees work in ML techniques. No reference in lines 56-58: “Not only can ML provide potentially more accurate predictive models, but techniques such as decision trees can provide new insights into non-linear relationships.”. 

RESPONSE: Given that decision tress are not one of our actual analytic models we have removed this from the introduction as to not confuse the reader.

2. Lines 82-83: “there are no studies delineating both the shared and unique risks” is the italic necessary? 

RESPONSE: We appreciate the question and agree that there does not need to be italics. It has been removed.

3. Why did CW=child welfare and non-CW turn into Line 117: maltreated and comparison groups? Please have uniformity. 

RESPONSE: We apologize for this inconsistency and have revised the language to be consistent.

4. What does a “biracial” participant’s race mean?

RESPONSE: We have updated this term to Multi-racial, meaning the participant reported 2 or more races/ethnicities.

5. According to data analysis, why do not present results before and after multiple imputations? The missingness proportion is less than 2%, but if you apply multiple imputations, results sure present how data and results look before imputation. 

RESPONSE: We thank the reviewer for raising this point. Although the variable level missingness is <2%, listwise deletion would results in 10% of the sample being dropped form the analyses. As such we feel that is it critical to use imputation to mitigate systematic bias due to missingness. 

We have completed the analyses for the data with missingness and report that there is no difference in the model performance metrics between the imputed and non-imputed data. As such we report only the results for the feature importance and coefficients for the imputed data. We include the performance metrics for the non-imputed data in a supplementary table.

6. Here is one serious concern with the analytical approach: Lines 242-243 describe how “generally a value higher than 0.7 designates a good model, and higher than 0.8 a strong model”. Table 3 presents AUC for all models between 0.79 and 0.83. That means all models (i.e., Logistic Regression, Lasso and SVM) are good or strong. Why do you opt for different models under small and marginal differences? 

RESPONSE: Although the difference in the AUC may seem small, any incremental increase in accuracy is considered an improvement. However, in our revised analyses with the new race/ethnicity variables we found that the performance metrics across the three models were so similar we could not choose a best-fitting model. Therefore, we now report the feature selection and coefficients across all three models. This adds robustness to our results as we only retained the features that met our threshold across all three models. This mitigates differences in features selected by each model that may be due to model-specific algorithms.

7. Following the previous comment in Line 237- 239: In linear SVM, one can interpret the magnitude and sign of the coefficient in the linear hyperplane similarly to the coefficients in Logistic Regression and Lasso. What references do you have for this affirmation? 

RESPONSE: In those models the magnitude of the coefficient means the same thing for the prediction of the outcome. We have added the following reference: Rakotomamonjy, A. (2003). Variable selection using SVM-based criteria. Journal of machine learning research, 3(Mar), 1357-1370.

8. Also, it is unclear whether your study is a comparison (e.g., CW=child welfare and non-CW), but you might use non-comparable models (e.g., Logistic Regression vs SVM). Unusually, two different models are recognized for two independent samples under the same analysis. For CW, it is SVM, and for Non-CW, it is the Logistic Regression. It is necessary to have a better explanation and justify with technical background why a model may be more applicable to one group and not to the other. This point needs a deep rationale for your data analysis proposal. 

RESPONSE: In our revised analyses we present all three models for both CW and non-CW youth.

9. Line 272 It is “perturbed” or “permuted”? 

RESPONSE: We apologize for this error and have changed it to permutated.

10. Why do you use the restrictions of the Lasso model and the technique called Backward Feature Selection? It seems like there are too many restrictions for the exploration of Line 46: “further work to delineate the relative importance of known predictors of substance use /…/”. 

RESPONSE: Lasso does not have restrictions, regularization is used to overcome issues with over fitting when many features are used that may duplicate variance. This is an advantage of Lasso over logistic regression. Backward feature selection is a distinct part of the process in which each feature is iteratively removed and the predictive performance with a single feature removed is evaluated and the final set of features is chosen based on the set that results in the best AUC.

11. How sensitive is the Permutation Feature Importance (PFI) to the SVM and the logistic regression results?

RESPONSE: Permutation feature importance is model agnostic as stated on line 307-308.

12. It is not easy to interpret the values from the Permutation Feature Importance (PFI) analysis. However, if the arbitrary selection of models we asked about in point 7 is justifiable, at least all results in all models should be presented for a fair comparison.

RESPONSE: For each model a higher value means the feature is contributing more to the prediction of the outcome. It can be approximated to mean the amount the predictive accuracy (AUC) will drop if that feature is removed for the model. We stated in the text that we used a cut-off of .005 to report important features, which means a contribution of ≥.5% to the accuracy. In the results section we note that the top feature for CW youth is peer marijuana use which would decrease the AUC by 12-13% if dropped from the model. Similarly, for non-CW youth externalizing behavior is the top feature and would result in a drop of 10-12% if removed from the model. These clarifications have been added to the analysis and results sections.

13. Since the authors claim that this is a study “Line 396-398: to provide evidence regarding the comparability of risk factors for substance use among CW-involved versus non-CW-involved youth, but the first to so within one study and using machine learning approaches.”, it is capital to clarify the analytic concerns mentioned before. The richness of their approach will reside in how well the points 5 to 11 are corrected and rationalized. 

RESPONSE: We appreciate the opportunity to clarify our analytic methods in the above points.

---

## [Decision Letter · Decision Letter 1]

9 Sep 2022

Using machine learning to determine the shared and unique risk factors for marijuana use among child-welfare versus community adolescents

PONE-D-21-15650R1

Dear Dr. Negriff,

We’re pleased to inform you that your manuscript has been judged scientifically suitable for publication and will be formally accepted for publication once it meets all outstanding technical requirements.

Kind regards,

Carlos Andres Trujillo, PhD

Academic Editor

PLOS ONE

Additional Editor Comments (optional):

Reviewers' comments:

Reviewer's Responses to Questions

**Comments to the Author**

1. If the authors have adequately addressed your comments raised in a previous round of review and you feel that this manuscript is now acceptable for publication, you may indicate that here to bypass the “Comments to the Author” section, enter your conflict of interest statement in the “Confidential to Editor” section, and submit your "Accept" recommendation.

Reviewer #1: All comments have been addressed

Reviewer #2: All comments have been addressed

2. Is the manuscript technically sound, and do the data support the conclusions?

Reviewer #1: Yes

Reviewer #2: Yes

3. Has the statistical analysis been performed appropriately and rigorously? 

Reviewer #1: Yes

Reviewer #2: Yes

4. Have the authors made all data underlying the findings in their manuscript fully available?

Reviewer #1: Yes

Reviewer #2: No

5. Is the manuscript presented in an intelligible fashion and written in standard English?

Reviewer #1: Yes

Reviewer #2: Yes

6. Review Comments to the Author

Reviewer #1: I think the authors have adressed all my concerns. I think it has improved and is ready for publication. I really like the research.

Reviewer #2: I appreciate the effort and clarity of the authors in every answer to the comments. Also, I have no additional comments for the authors or concerns about dual publication, research, or publication ethics.

7. PLOS authors have the option to publish the peer review history of their article (what does this mean?). If published, this will include your full peer review and any attached files.

Reviewer #1: **Yes: **Angela Trujillo

Reviewer #2: **Yes: **Juan Jose Giraldo-Huertas

---

## [Editor Report · Acceptance letter]

13 Sep 2022

PONE-D-21-15650R1 

Using machine learning to determine the shared and unique risk factors for marijuana use among child-welfare versus community adolescents 

Dear Dr. Negriff:

I'm pleased to inform you that your manuscript has been deemed suitable for publication in PLOS ONE. Congratulations! Your manuscript is now with our production department. 

Kind regards, 

on behalf of

Dr. Carlos Andres Trujillo 

Academic Editor

PLOS ONE